# Working at Green Care Farms and Other Innovative Small-Scale Long-Term Dementia Care Facilities Requires Different Competencies of Care Staff

**DOI:** 10.3390/ijerph182010747

**Published:** 2021-10-13

**Authors:** Bram de Boer, Yvette Buist, Simone R. de Bruin, Ramona Backhaus, Hilde Verbeek

**Affiliations:** 1Department of Health Services Research, Care and Public Health Research Institute, Maastricht University, 6200 MD Maastricht, The Netherlands; r.backhaus@maastrichtuniversity.nl; 2Centre for Nutrition, Prevention and Health Services, National Institute for Public Health and the Environment (RIVM), 3720 BA Bilthoven, The Netherlands; yvette.buist@wur.nl (Y.B.); sr.de.bruin@windesheim.nl (S.R.d.B.); 3Social Sciences Department, Chairgroup Health and Society, Wageningen University & Research, 6708 PB Wageningen, The Netherlands; 4Research Group Living Well with Dementia, Department of Health and Wellbeing, Windesheim University of Applied Sciences, 8017 CA Zwolle, The Netherlands

**Keywords:** long-term care, dementia, care environment, competencies, innovative design

## Abstract

The culture change movement within long-term care in which radical changes in the physical, social and organizational care environments are being implemented provides opportunities for the development of innovative long-term care facilities. The aim of this study was to investigate which competencies care staff working at green care farms and other innovative types of small-scale long-term dementia care facilities require, according to care staff themselves and managers, and how these competencies were different from those of care staff working in more traditional large-scale long-term dementia care facilities. A qualitative descriptive research design was used. Interviews were conducted with care staff (*n* = 19) and managers (*n* = 23) across a diverse range of long-term facilities. Thematic content analysis was used. Two competencies were mainly mentioned by participants working in green care farms: (1) being able to integrate activities for residents into daily practice, and (2) being able to undertake multiple responsibilities. Two other competencies for working in long-term dementia care in general were identified: (3) having good communication skills, and (4) being able to provide medical and direct care activities. This study found unique competencies at green care farms, showing that providing care in innovative long-term care facilities requires looking further than the physical environment and the design of a care facility; it is crucial to look at the role of care staff and the competencies they require.

## 1. Introduction

In many countries, there is a movement towards developing age-friendly cities and environments [1,2,3]. Environments should be inclusive, accessible and optimize opportunities for participation and security for all people, including older people, and people with dementia. As people with dementia become more dependent on their environment as the dementia progresses, careful design of the physical space they reside in has important implications for maintaining a meaningful life. As the environment can provide support in dealing with cognitive decline, the importance of a good fit between the person with dementia and their environment is crucial.

As part of the movement towards age-friendly cities, there is an increased development of alternative care environments for people with dementia. In addition, recent insights show that it is important that changes in the physical environment of a nursing home are made in conjunction with changes in the social environment (e.g., way of working and required competencies) [1,4,5]. Traditional care environments for people with dementia are often closed, separated from the community, with a focus on keeping residents safe and confined [6,7]. In contrast, innovative care environments, such as green care farms, aim to promote the health and well-being of people with dementia by following design principles aimed at, for instance, optimizing stimulation, supporting engagement, and creating a link with the community [5,8,9,10,11]. Green care farms are mostly situated in rural environments and combine agricultural activities with care activities [12]. These green care farms focus on the interrelatedness of the physical, social and organizational environments. This means that working in green care farms may require different competencies of care staff compared with regular long-term care facilities [13,14,15,16]. Green care farms have a unique physical environment, and care staff are expected to make optimal use of this environment by facilitating activity engagement and autonomy, and focusing on maintaining skills rather than on lost abilities [16,17,18,19]. The physical environment offers many opportunities (e.g., presence of animals, plants, and natural aspects) for activity engagement. Care staff play an important role in incorporating activities into normal daily care practice, yet they should have the competencies to do so.

A competency can be defined as a cluster of related knowledge, attitudes, and skills that affect a major part of one’s job [20]. In general, prior research indicates that a focus on numbers (e.g., staffing levels) does not necessarily lead to improvements in quality of care. Therefore, more research is needed paying attention to the required competencies of care staff [21,22], especially in long-term care environments. An integrative review on which competencies of licensed practical and registered nurses are needed for older people nursing identified five competence areas [23]. The identified competencies concerned the ability to ensure older peoples’ dignity and quality of life; interacting with residents, family and professionals; evidence-based practice; supervision; and leadership. In addition, an initial study by van Stenis et al. (2017) identified a list of competencies that were deemed necessary for caregivers in order to fulfill their changing role in nursing homes effectively. This list included competencies related to communication, attentiveness, negotiation, flexibility, teamwork, expertise, and coaching/leadership [24]. However, these competencies are general desired competencies, mainly nursing specific, and it is unclear whether they apply to all types of long-term care environments. For example, studies have reported different job characteristics for nursing staff in small-scale, homelike work environments, including more perceived job autonomy (including decision-making authority) and social support [25,26]. In addition, the role of urban green spaces inside and outside care facilities for older people is increasing across European cities as well [27,28,29]. These developments highlight the importance of investigating which competencies of care staff are required. As there is no ‘one size fits all’ approach to employing care staff, it is important to consider differences between care environments.

Thus, more insight is needed into which competencies are considered important for care staff according to care staff themselves and managers, and how these competencies may differ across different types of long-term care facilities [30]. It is important to consider both the care staff and management perspectives, as managers tend to have a broader view of staff roles within their organization, whereas care staff are more likely to report on their actual experiences during care delivery [21]. Therefore, the aim of this study was to investigate which competencies care staff working at green care farms and other innovative types of small-scale long-term dementia care facilities require according to care staff themselves and managers, and how these competencies were different from those of care staff working in more traditional large-scale long-term dementia care facilities.

## 2. Materials and Methods

### 2.1. Design

A qualitative, descriptive research design [31] was used in which two data sources from earlier studies were used [32,33]. Both data sources included data that were collected in three types of long-term care facilities: green care farms, small-scale care facilities, and large-scale care facilities. Table 1 provides a brief description of these three types of long-term care facilities. Data source 1 consisted of interviews conducted between April and September 2014. It focused on the care staff perspective and was aimed at describing which competencies, according to the care staff, are required for working in a particular type of nursing home. Data source 2 consisted of interviews conducted between March and July 2016. It took a broader perspective and focused on the management perspective. The aim was to describe which competencies, according to managers, are required for working in a particular type of long-term care facility, and included facilities providing adult day services, nursing home care, or both.

### 2.2. Sample/Participants

The respondents in both data sources were purposefully selected [34] in order to make sure that respondents from green care farms, small-scale care facilities, and large-scale care facilities were included. Both the care staff and management perspectives were considered, as a previous study showed that there can be discrepancies between these perspectives [35]. Participants (care staff and managers) were invited by email or telephone to participate in the studies. In data source 1, care staff (i.e., nursing aids, nursing assistants, certified nursing assistants and registered nurses) directly involved in care tasks and working on a permanent basis in either the selected small-scale living facilities or regular psychogeriatric wards were eligible to participate in the study. Temporary staff (such as trainees), permanent night-shift workers and team managers were excluded from the study. In data source 2, participants were eligible to participate when they had a leading function (e.g., manager, team leader or coordinator) and conducted overarching tasks related to hiring and coaching employees. In both initial studies, the principle of data saturation was used [36], meaning that interviews were conducted until no new themes emerged and there was a high rate of recurrence of responses.

### 2.3. Data Collection

Based on geographic convenience, each interview was conducted by either the first or second author, or a trained research assistant (three interviewers in total). All interviewers had previous experience with conducting semi-structured interviews. The interviewers visited the respondents at their care facilities and followed a semi-structured interview format, which was tailored to the type of care facility the respondent worked in. Interviews took place in a quiet room of the care facility and lasted about one hour. All interviews were audio recorded and transcribed verbatim. Table 2 gives an overview of the relevant interview topics, including example questions.

### 2.4. Data Analysis

Data were analyzed using MAXQDA 12 [37]. Since the aim of the study described in this paper was different from the aims of the studies from which the data were initially collected, we considered our approach to be a secondary data-analysis. The data-analysis process is shown in Figure 1. First, two researchers (Authors 1 and 2) familiarized themselves with the data by reading the original transcripts. Second, topics from the topic list were used as a basis for the initial codes, and text passages that could not be coded with these initial codes were given a new code describing the content of the passage [38]. Third, the coding tree that emerged from the first several interviews was used for coding all interviews. All interviews were divided between the two researchers. Each researcher coded half of the interviews, and checked the codes of the other half. In the case of disagreement, codes were discussed within the research team to reach agreement. After coding, recurring themes were defined by clustering initial codes into overarching categories. We continuously compared the views of care staff and management during analysis to describe possible variation between these perspectives [39]. Lastly, drafts of the study findings were developed by identifying core themes and variation across the types of nursing homes and perspectives. After all interviews were analyzed, the research team concluded that data saturation was reached, as no new themes emerged in the data.

By having two researchers independently coding a subset of the interviews, and checking the codes of the other half, the trustworthiness of the analysis was increased [31]. Furthermore, an iterative approach was used, meaning that during the data analysis process, the research team repeatedly went back to the original transcripts in order to verify their conclusions, which contributed to the credibility of the findings. In addition, additional strategies for ensuring the rigor of data were applied [40]. Qualitative analysis software for systematic analysis was used. Furthermore, we applied the principle of space triangulation within the different types of nursing homes (by collecting data at multiple sites, to test for cross-site consistency) and person triangulation, which entails the collection of data from different respondent groups [31].

## 3. Results

In total, the data sources included 42 interviews that were held within 30 different long-term care facilities. Nineteen interviews were conducted with care staff; six participants at two green care farms; seven at three small-scale care facilities; and six at two large-scale care facilities. The majority of the participants were certified nursing assistants (two to three years education in the Netherlands). Twenty-three interviews (within twenty-three different long-term care facilities) were conducted with managers, team leaders, or coordinators; 12 at green care farms; five at small-scale care facilities; and six at large-scale care facilities.

Two competencies were mainly mentioned by participants working in green care farms (and partly in other small-scale facilities): (1) being able to integrate activities for residents into daily practice, and (2) working independently. Two other competencies for working in long-term dementia care in general were identified: (3) having good communication skills, and (4) being able to provide medical and direct care activities. Table 3 shows the competencies that were identified as being important according to either care staff or management within the different facilities.

### 3.1. Being Able to Integrate Activities for Residents into Daily Practice

Managers and care staff of green care farms consistently mentioned that ‘being able to integrate activities for residents into daily practice’ was an important competency for care staff. At green care farms, care staff talked about the fact that besides taking care of people with dementia, they formed a household together with them, meaning, for example, that they also cook and clean together.

‘This is different than in traditional nursing homes, at green care farms we do everything. We are responsible for the actual caregiving, but also for counseling, providing activities, cooking, and cleaning, the whole package’.(care staff member green care farm 3)

Care staff at green care farms also indicated that, in order to integrate activities for residents into normal daily practice, a certain amount of flexibility is required. They indicated that they tried to be flexible in order to include residents in the activities as much as possible.

‘When I have to clean a resident’s room resident, I always try to take the resident with me. I like that, because it can be fun and cozy, we can make an activity out of it, and have some fun for 15 min. Sometimes they like that, and then I ask them to help me with cleaning. This is also a sign of respect I think, because I am in their private space, touching their stuff, so it makes sense to include them in the activity. I am in ‘their house’ after all’.(care staff green care farm 1)

In contrast, care staff and managers in large-scale environments did not refer to this competency at all. At some of the small-scale facilities, care staff and management talked briefly about the fact that they had to integrate activities for residents into daily care, and had to carry out activities together with residents. They talked about the fact that staff have to be able to multitask, hence also being responsible for more than just taking care of people with dementia.

‘Staff is responsible for multiple tasks at once. For instance, you are responsible for cooking, but at the same time there is a group of people that you have to supervise. Additionally, you have to be able to do both, so you have to make sure you do not cook the potatoes too long, and at the same time keep an eye out for residents wandering off. Those skills are really needed here, and if you think that is difficult, than we have a problem’.(manager small-scale facility 5)

### 3.2. Being Able to Undertake Multiple Responsibilities

The competency to bear multiple responsibilities was mentioned consistently in green care farms and by about half of the respondents in other small-scale, homelike care facilities. In contrast, in large-scale facilities, this was not mentioned at all. Care staff at green care farms and small-scale facilities mentioned that they worked alone for most of the day and that they needed to make decisions regarding care, activities and the daily lives of the people with dementia they were responsible for. Asking a colleague would mean the group had to be left alone. This meant that care staff would need to be able to undertake these responsibilities by themselves.

‘It is just you alone working on a group. You have to decide what the day will look like. There is no supervisor or manager to tell you what to do. Now, you have to make those judgements yourself’.(manager green care farm 4)

As opposed to working independently, care staff in large-scale facilities mentioned that it was important to be able to work with others. However, working with others was often described as task differentiation or dividing responsibilities, meaning that each staff member was made responsible for a specific task, and therefore, collaboration is important to complete all activities.

‘We are working together all day, we complement each other. We all have our own qualities. I’m mostly busy with arranging things, walking rounds, reporting incidents, etc… and the nursing aid is busy with the basic care tasks…. The kitchen staff is also important. As I cannot do everything, then I would be overwhelmed’.(care staff large-scale facility 2)

### 3.3. Having Good Communication Skills

Having good communication skills was the most commonly mentioned competency across all facilities by care staff and management. There was wide agreement that in all facilities, it is important to tailor communication to a specific individual, and approach people with dementia in a respectful manner. Participants talked about empathy, being attentive, and having a warm personality when referring to good communication skills.

‘Being positive and happy, and having a respectful approach. Always take people seriously, no matter the stage of dementia they are in, because their feelings do not go away. Additionally, if you then approach them as if they are a child, or if you are being disrespectful, or you do not include them in a conversation at all… they feel that as well’.(manager small-scale facility 10)

Furthermore, being able to communicate with family members of people with dementia was also mentioned. Taking into account the perspectives and experiences of family members when talking to them was identified as being an important competency as well.

### 3.4. Being Able to Provide Medical and Direct Care Activities

Managers in all facilities agreed that ‘being able to provide medical and direct care activities’ is one of the basic competencies that all staff should possess. Managers mentioned that the care staff need to have a minimum level of knowledge, which enables them to provide medical and direct care activities to all people with dementia, and medical competencies, such as being able to provide medication, handling wound care, and taking care of people with stomata. In particular, managers at green care farms mentioned that medical expertise should be kept up to date with regular trainings, either through official courses or through informal gatherings with staff members.

‘We arrange evenings in which all staff gather, for instance to talk about lifting techniques, or dealing with medication. Furthermore, staff members can follow courses on for instance medication use, etc. we want to expand these opportunities for staff in the future’.(manager green care farm 8)

Care staff at large-scale facilities also referred to this as an important competency; care staff working in small-scale living facilities recognized it to some extent. In contrast, the competency was not explicitly mentioned as being important by care staff at green care farms.

## 4. Discussion

This study aimed to investigate which competencies care staff working at green care farms and other innovative types of small-scale long-term dementia care facilities require according to care staff themselves and managers, and whether these competencies differ from those of care staff working in more traditional large-scale long-term dementia care facilities. Staff and managers from greens care farms emphasized the importance of being able to integrate activities for residents into daily practice, and being able to undertake multiple responsibilities. These competencies were not mentioned at large-scale more traditional facilities, and were acknowledged to a lesser extent by staff and managers from other small-scale facilities. At regular settings, there was more emphasis on ‘being able to provide medical and direct care activities’. In general, having good communication skills was considered important, regardless of setting and perspective (care staff or management).

Variation between facilities was found for care staff of green care farms and large-scale facilities, which could be traced back to the underlying care concept of these different facilities. Many large-scale facilities still emphasize a medical model of care, which might explain the fact that care staff in these facilities prioritized medical care tasks, and worked according to a strict task differentiation [16,41]. This could also be related to the hierarchical structure within large-scale facilities in which specific staff members are responsible for specific tasks. There is little integration of care tasks within the daily life of residents, which can lead to care staff having little impact on daily life at the facility (as this is not their responsibility). Care staff at green care farms focused on being able to integrate activities for residents into care practices and being able to work independently and having multiple responsibilities. This may partly be explained by the increased integration of direct care activities within facilities that provide person-centered care in a homelike atmosphere with an increased focus on well-being-related activities. In addition, it could be related to the interrelatedness of the physical, social and organizational environment of green care farms suggested in previous studies [4,18,33,42]. Having a clear vision of how the physical environment should be used, and how care staff should implement this, regardless of their function, is a unique characteristic of green care farms. The social environment contains the care staff, which is essential in facilitating an optimal use of the opportunities that the physical care environment of green care farms offer. In addition, organizational aspects, such as competencies of managers, are important as well (e.g., consultative, facilitative and flexible leadership; vision to implement a radically different care philosophy and to perform tasks differently).

These findings are in line with previous studies that suggested that care staff working in innovative, small-scale and homelike facilities for older people with dementia have different roles and tasks compared with care staff in traditional facilities [43,44]. Sharkey et al. (2011), for instance, showed that the Green House model leads to expanded responsibilities of nursing staff, and more time spent on active engagement with residents [45]. However, there is a large variation in care practices by nursing staff within different ‘Green House’ facilities [46,47]. Furthermore, care staff in green care farms are directed to focus more on remaining capacities and strengths of people with dementia compared to other long-term care facilities [18]. In addition, a previous study on competencies that were deemed necessary for caregivers to fulfill their changing roles in nursing homes stated that the role of care staff is shifting from a task-oriented approach towards a relation-oriented approach, which comprises skills and competencies that concern communication; attentiveness; negotiation; flexibility; teamwork; or expertise, coaching and leadership [24].

The finding that communication skills were considered to be important across all facilities and perspectives is not surprising. The role of communication in nursing care is well established [48,49]. In particular, for people with dementia, it is important to focus on aspects such as mutuality, autonomy, respect, and trust during communication [50].

Our study did not identify leadership as an important competency, in contrast with other studies [23,24,51]. One possible explanation for this could be that in our study, we focused on generic competencies that are needed during direct care activities, regardless of educational level. Coaching/leadership is more associated with indirect activities, and might be more relevant for higher educated nursing staff, such as baccalaureate-educated registered nurses [51,52]. In our study, we did not differentiate according to level of education. In general, only a few members of nursing staff with bachelor’s degrees are working in nursing homes in the Netherlands. Of all the care staff working in Dutch nursing homes, probably less than 5% are nurses with bachelor’s degrees, and specific numbers regarding types of setting are unknown. However, it is important that future studies focus on leadership behaviors in innovative long-term care facilities, as previous studies show that nursing home managers’ leadership is associated with, for instance, job strain [53].

Some implications can be drawn from this study. The required competencies demand changes in education and recruitment practices. Curricula for direct care staff should integrate teaching about competencies required at innovative care facilities in their education practices. Furthermore, during the recruitment of new care staff, especially large-scale, more traditional long-term care facilities should focus on recruiting nursing staff that possess competencies that are required to provide innovative, small-scale, and person-centered care. Future studies should investigate possibilities for facilities to aid care staff in their development of required competencies. In addition, rather than focusing on general competencies, these studies should aim to identify which specific knowledge, attitudes and skills are required.

Some methodologies should be considered. This study is based on interviews and, therefore, we do not have evidence on competencies of care professionals in real-life everyday care. Hence, the professionals might have given different answers to the questions on competencies compared to competencies that are actually used in practice. It is a strength that we included both the care staff and management perspectives in the different types of care facilities and locations (person + space triangulation). However, we cannot check whether personal characteristics, such as gender, age, or years of working experience, or work environment characteristics, such as staffing levels, had an influence on our findings, as the studies from which the data sets were collected did not gather this information from the respondents. Lastly, the use of secondary interview data of different cohorts and time periods impedes the generalizability of the findings.

## 5. Conclusions

This study shows that having good communication skills and being able to provide medical and direct care activities are competencies required in all long-term dementia care facilities. In addition, providing care in innovative long-term care facilities requires care staff to be able to integrate activities for residents into daily practice and to be able to undertake multiple responsibilities. Considering the changing role of care staff in nursing homes, large-scale, more traditional long-term care facilities should place more emphasis on competencies required for a psychosocial care approach.

## Figures and Tables

**Figure 1 ijerph-18-10747-f001:**
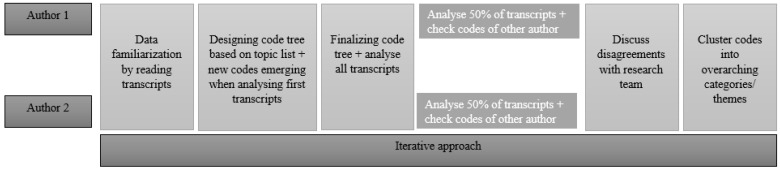
Data analysis process.

**Table 1 ijerph-18-10747-t001:** Description of long-term care facilities.

Facility	Description
Innovative concepts	
Green care farms	Green care farms combine agricultural activities with care and support services for people with dementia. Meaningful and stimulating activities, such as preparing meals, gardening, and taking care of animals, are encouraged in a unique physical environment. People with dementia are surrounded by nature, and are free to move as they please. Activities are integrated into normal daily care practices as much as possible.
Small-scale care facilities	Small-scale care facilities are facilities in a homelike environment that can be either stand-alone facilities or facilities clustered on the grounds of a larger-scale nursing home. As green care farms, small-scale facilities generally take an innovate approach to dementia care.
Traditional concepts	
Large-scale care facilities	Large-scale care facilities have a more institutional atmosphere. The routines are often determined by nursing staff, and there is a more medical model of care compared to green care farms and other small-scale facilities.

**Table 2 ijerph-18-10747-t002:** Interview overview.

	Interview Topic	Example Questions
Data source 1	Competencies Skills Knowledge Attitudes Differences between facilities	Which skills do you need to work in this facility? Does working in this facility require specific knowledge? Does working in this facility require a specific attitude? What do you believe, could be the main differences between this and other (small/large/green care farm) facilities, and what could this mean for staff working at these facilities?
Data source 2	Competencies	Which competencies are, according to you, important for working in this facility? Is there any specific knowledge required? Do you need a particular attitude? Which skills are important?

**Table 3 ijerph-18-10747-t003:** Identified competencies and differences between facilities and perspectives.

	Green Care Farm		Small-Scale Facility		Large-Scale Facility	
Competency	CS	M	CS	M	CS	M
Being able to integrate activities for residents into care practice						
Being able to undertake multiple responsibilities						
Having good communication skills						
Being able to provide medical and direct care activities						

CS = care staff; M = management. Green cells represent little variation within a group of respondents regarding indicating the competency as being important (all agreed on the importance). Orange cells represent discrepancy within a group of respondents. Red cells indicate that no respondent indicated the competency as important.

## Data Availability

The data presented in this study are available on request from the corresponding author.

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
