# Peer review of "Working at Green Care Farms and Other Innovative Small-Scale Long-Term Dementia Care Facilities Requires Different Competencies of Care Staff"

_ijerph, 2021, doi:10.3390/ijerph182010747_

Round 1

Reviewer 1 Report

Thank you for inviting me to review this manuscript. Competence requirement for residential long-term care is an important topic. However, I have some comments which I think should be addressed prior publication.

Reviewer 2 Report

Dear authors, I first want to congratulate you for the theme addressed in this work, which can greatly contribute to health professionals

Title: The authors have a very long title, I suggest that it be rewritten in a more objective way and that it actually reflects the development of the work. “Working in a long-term care facility with an innovative design requires different skills from the care team? Perspectives of the service team and management"

Reviewer 3 Report

This is an interesting research, using a qualitative design. Results are potentially useful, since identify competences to care people in nursing homes at green care farms.

Strength, a novelty to look for competences was explored by using an alternative care environment. Results are potentially useful, since they identify competences to care for people with dementia in nursing homes.
